# A solar-driven atmospheric water extractor for off-grid freshwater generation and irrigation

Kaijie Yang [1,2,3,7], Tingting Pan[1,7], Nadia Farhat[3], Alejandra Ibarra Felix [3], Rebekah E. Waller[4], Pei-Ying Hong [3], Johannes S. Vrouwenvelder [3], Qiaoqiang Gan [2,3] ✉ & Yu Han [1,5,6] ✉

Solar-driven atmospheric water extraction (SAWE) is a sustainable technology for decentralized freshwater supply. However, most SAWE systems produce water intermittently due to the cyclic nature, with adoption hindered by complex design requirements or periodic manual operations. Herein, a fully passive SAWE system that can continuously produce freshwater under sunlight is presented. By optimizing the three-dimensional architecture to facilitate spontaneous mass transport and efficient energy utilization, this system can consistently produce $0.65 \, \text{L m}^{-2} \text{h}^{-1}$ of freshwater under 1-sun illumination at 90% relative humidity (RH) and functions in arid environments with an RH as low as 40%. We test the practical performance of a scaled-up system in Thuwal, Saudi Arabia over 35 days across two seasons. The system produces $2.0–3.0 \, \text{L m}^{-2}$ per day of freshwater during the summer and $1.0–2.8 \, \text{L m}^{-2}$ per day of freshwater during the fall, without requiring additional maintenance. Intriguingly, we demonstrate the system's potential for off-grid irrigation by successfully growing cabbage plants using atmospheric water. This passive SAWE system, harnessing solar energy to continuously extract moisture from air for drinking and irrigation, offers a promising solution to address the intertwined challenges of energy, water, and food supply, particularly for remote and water-scarce regions.

Freshwater scarcity is a paramount global challenge, impacting 2.2 billion people[1,2]. This issue affects not only those residing in arid and remote regions where transporting water over long distances is either costly or impractical[3], but also developed island territories or coastal regions with no stable freshwater resources[4]. Furthermore, this situation has been exacerbated by climate change and rapid population growth[5]. Agricultural irrigation and electrical power generation are the two primary processes requiring freshwater, accounting for 70% and 15% of global freshwater withdrawals[6,7], respectively. However, freshwater production relies on energy input[6], underscoring the vital role of the water–energy–food nexus in achieving sustainable development[8,9].

[1]Advanced Membranes and Porous Materials Center, Physical Sciences and Engineering Division, King Abdullah University of Science and Technology, Thuwal 23955-6900, Saudi Arabia. [2]Sustainable Photonics Energy Research Lab, Material Science Engineering, Physical Science and Engineering Division, King Abdullah University of Science and Technology, Thuwal 23955-6900, Saudi Arabia. [3]Water Desalination and Reuse Center, Division of Biological Sciences and Engineering, King Abdullah University of Science and Technology, Thuwal 23955-6900, Saudi Arabia. [4]Center for Desert Agriculture, Division of Biological Sciences and Engineering, King Abdullah University of Science and Technology, Thuwal 23955-6900, Saudi Arabia. [5]Center for Electron Microscopy, South China University of Technology, Guangzhou 511442, China. [6]School of Emergent Soft Matter, South China University of Technology, Guangzhou 511442, China. [7]These authors contributed equally: Kaijie Yang, Tingting Pan. ✉e-mail: qiaoqiang.gan@kaust.edu.sa; yu.han@kaust.edu.sa

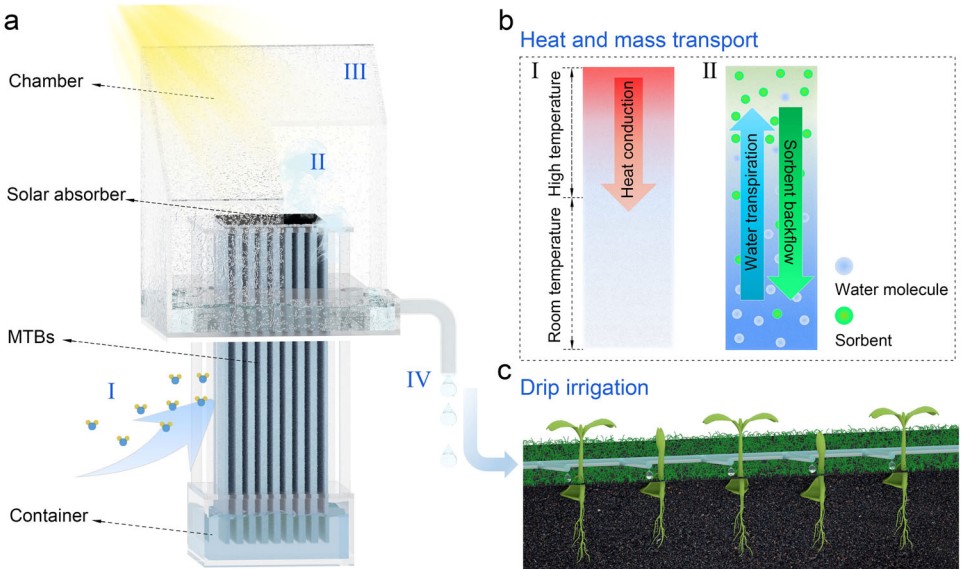

**Fig. 1 | System design and working principle. a** Schematic of the system architecture and freshwater production process, which includes (I) atmospheric water capture by the lower part of the MTBs structure exposed to the environment, (II) solar-driven vapor generation from the upper part of the MTBs structure enclosed within a chamber, (III) vapor condensation on the chamber's inner surface, and (IV) freshwater collection from the condensation. **b** Illustration of heat and mass transport. (I): Heat generated by the solar absorber is conducted downward along the MTBs. (II): As the vapor is released, the sorbents are driven downward along the MTBs due to the concentration gradient, whereas the captured water molecules are transpired, traveling upward along the MTBs. **c** Illustration of off-grid irrigation by directing the produced water (from **a**) to the roots of plants.

The use of renewable energy sources for freshwater production, for both drinking and irrigation, is essential to meet the increasing demands for water, energy, and food. Atmospheric water, a ubiquitous natural resource, is estimated to contain six times the total freshwater volume in rivers across the globe[3,10]. Moreover, its availability is projected to increase due to global warming[11,12]. Solar-driven atmospheric water extraction (SAWE) has emerged as a highly promising method for decentralized freshwater supply, particularly in remote, water-scarce regions[13–16]. Typically, SAWE systems use hygroscopic sorbents to capture moisture from the surrounding environment[16,17]. Upon reaching saturation, the system is sealed and exposed to sunlight to initiate the release of captured water. This process enables SAWE to transcend geographical and climatic constraints, rendering it a more universally applicable technology compared to other passive atmospheric water extraction technologies such as fog and dew collection[15,16,18].

However, previous SAWE systems often faced limitations due to the slow kinetics of sorbent materials, allowing only one sorption–desorption cycle per day, with moisture capture at night and water production during the day[19,20]. Thus, the productivity of these systems is inherently constrained by the sorbent adsorption capacity. This limitation can be overcome using multicycle systems by developing sorbents or sorbent beds with rapid kinetics[21–23]. Despite promising advancements, the widespread adoption of this technology remains constrained by the high cost of nanomaterials and challenges associated with scaling up prototypes[24,25]. Moreover, due to their cyclic nature, these systems can only produce water intermittently under sunlight[26] and switching cycles requires either an active system or labor-intensive operation and auxiliary moving parts, resulting in an energy-intensive process and system complexity[23]. To fully harness the immense potential of SAWE, a truly passive and scalable system that can efficiently produce freshwater without labor-intensive maintenance is required.

In this study, we present a stand-alone SAWE system capable of producing freshwater without requiring maintenance, solely using sunlight. Under 1-sun illumination, the system produces 0.65 L m$^{-2}$ h$^{-1}$ of freshwater at 90% relative humidity (RH) and functions continuously in environments with an RH as low as 40%. The global potential of the system was assessed based on solar irradiation and humidity distribution. Results revealed that it is feasible in most areas and particularly well suited for equatorial regions with abundant solar irradiance and high humidity, achieving a maximum water production potential of 4.6 L m$^{-2}$ per day. A scaled-up version of the system was constructed that maintained the simplicity and affordability of the prototype design, and its performance was tested in Saudi Arabia. The system operated consistently under various seasonal weather conditions without requiring operation, attaining an optimal water productivity of approximately 3.0 L m$^{-2}$ per day. The atmospherically harvested water was successfully used for the off-grid irrigation of *Brassica rapa* (Chinese cabbage), highlighting its potential for point-of-use, off-grid irrigation in areas lacking access to large-scale water sources.

## Results

### System design and farbication

Figure 1a illustrates the design of the proposed system, underlining its key feature—the mass transport bridges (MTBs) structure, which is composed of numerous vertically aligned microchannels infused with a salt solution that serves as a liquid sorbent. Depending on the temperature distribution (Fig. 1b-I), the MTBs structure is divided into two functional regions: the room-temperature region exposed to environment for continuous atmospheric water capture (process I in Fig. 1a) and the high-temperature region enclosed in the chamber for freshwater generation (processes II–IV in Fig. 1a). During its operation, the room-temperature region captures atmospheric water and stores it in a container. When the system receives sunlight, the solar absorber converts the light into heat and generates concentrated vapor in the high-temperature region. The released vapor condenses on the chamber wall, producing freshwater. As water production proceeds, the captured water stored in the container transpires to the high-temperature region, ensuring uninterrupted and efficient vapor generation. Concurrently, the concentrated liquid sorbent in the high-temperature region is transported back to the room-temperature region via diffusion and convection, enabling

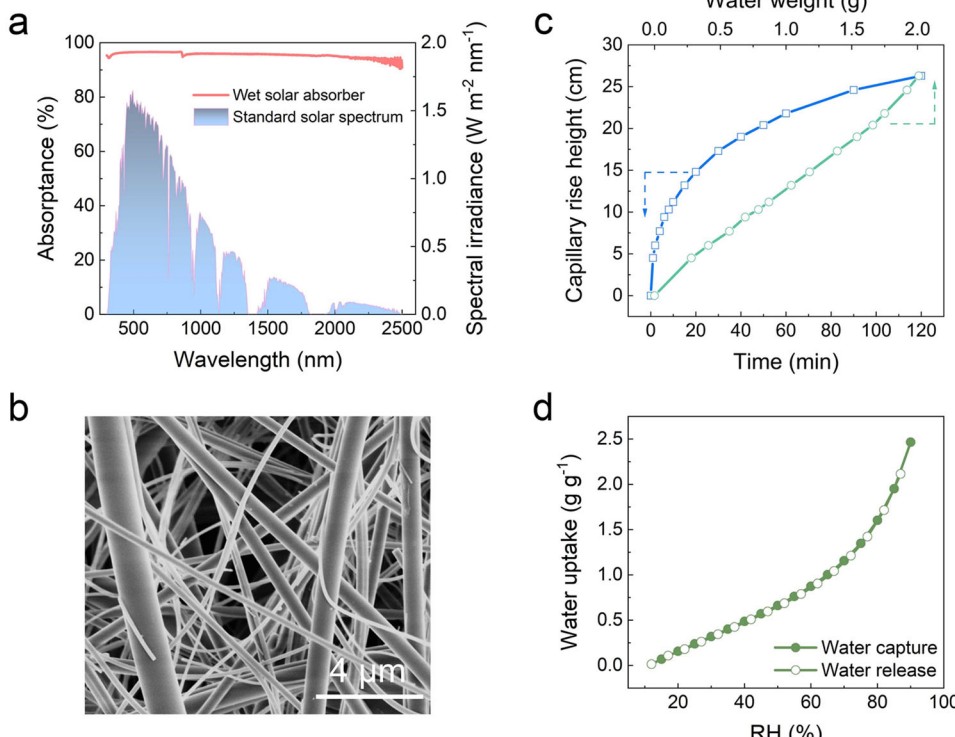

**Fig. 2 | Material selection for the system fabrication. a** UV–Vis–NIR spectrum of the solar absorber in the wet state, exhibiting a light absorptance of ~96%, in conjugation with the standard solar spectral irradiance (AM 1.5 G). **b** SEM image of GFM, showing the interwoven fibrous structure. **c** The blue curve illustrates the increase in capillary rise height of water within the GFM over time (see the recorded photos in Supplementary Fig. 2). The green curve depicts the linear escalation in the mass of water confined within the GFM, corresponding to the increase in capillary rise height. **d** Water sorption isotherm of the saturated LiCl solution.

it to continue capturing atmospheric water (Fig. 1b-II). The specially designed MTBs structure ensures highly efficient transport of water, sorbent, and utilization of heat throughout the entire process, thereby ensuring completely passive and maintenance-free atmospheric water production. By directing the condensed droplets to the plant roots, this device can facilitate off-grid irrigation using only atmospheric water (Fig. 1c).

To implement the proposed system, a solar absorber was fabricated by loading partially oxidized carbon nanotubes (CNTs) onto the glass fiber membrane (GFM). Owing to the light-trapping microstructures (Supplementary Fig. 1) and the inherent black property of CNTs, the solar absorptance of the resulting solar absorber can reach ~96% in the wet state (Fig. 2a). The GFM was used for fabricating MTBs owing to its hydrophilicity and intertwined fibrous structure (Fig. 2b). At the microscale, the intertwined fibers form abundant capillary microchannels, endowing the GFM with a strong water transport capability. The GFM can rapidly transport water (Fig. 2c), raising it to a height of over 10 cm within 10 min and reaching a height of approximately 27 cm after 2 h (Supplementary Fig. 2). Notably, the water uptake increased linearly with the height of capillary rise (Fig. 2c). This finding suggests that nearly all the microchannels are saturated with water, and thus, the interconnected, water-filled channels provide a path for the backflow of sorbents. Herein, a lithium chloride (LiCl) solution was selected as the hygroscopic liquid sorbent due to its availability, cost-effectiveness, wide range of applicable RH, and powerful water molecule capture capability[27,28]. As shown in Fig. 2d, the saturated LiCl solution can capture water molecules at an RH as low as 15% and the adsorption capacity reaches ~2.5 g g⁻¹ when the RH increases to 90%. Moreover, the water trapped by the LiCl solution can be efficiently released without hysteresis as the RH decreases. By assembling the solar absorber and the GFM sheets with a PMMA frame and

infiltrating the LiCl solution into the MTBs structure, a feasibility-verification prototype was developed with an evaporation area of 3 cm × 3 cm (Supplementary Fig. 3).

## Performance evaluation

The SAWE performance of the prototype was first tested in a controlled environment. When exposed to solar radiation, the solar absorber at the top of the MTBs converts the incoming light into heat energy. This energy is then conducted along the MTBs, creating a temperature gradient. Thus, the MTBs structure can be divided into two distinct functional zones (Supplementary Fig. 4): a vapor generation zone with high temperature (i.e., the upper MTBs structure enclosed in the chamber) and an atmospheric water capture zone at room temperature (i.e., the lower MTBs structure exposed to the environment). The MTBs structure was systematically adjusted to optimize the water production efficiency. The optimal heights for the vapor generation zone and atmospheric water capture zone were 3 and 5 cm, respectively, to maximize photothermal energy utilization and ensure sufficient moisture capture kinetics. Furthermore, 32 GFM sheets were used to construct the MTBs structure, guaranteeing efficient sorbent and water transport (see detailed optimization procedures in Supplementary Notes 1 and Supplementary Figs. 5–20).

A video was recorded to provide a visual insights into the real-time operation of the optimized prototype system with the MTBs structure infiltrated with 0.24 g g⁻¹ of LiCl solution at 90% RH (Supplementary Movie 1). In the absence of sunlight, the prototype system engaged in atmospheric water capture, storing the collected water in a container. As the process continued, the water level in the container increased (Fig. 3a-I). Upon exposure to sunlight, vapor was released and condensed to produce water (Fig. 3a-II). Simultaneously, a portion of the stored water was transported upward to

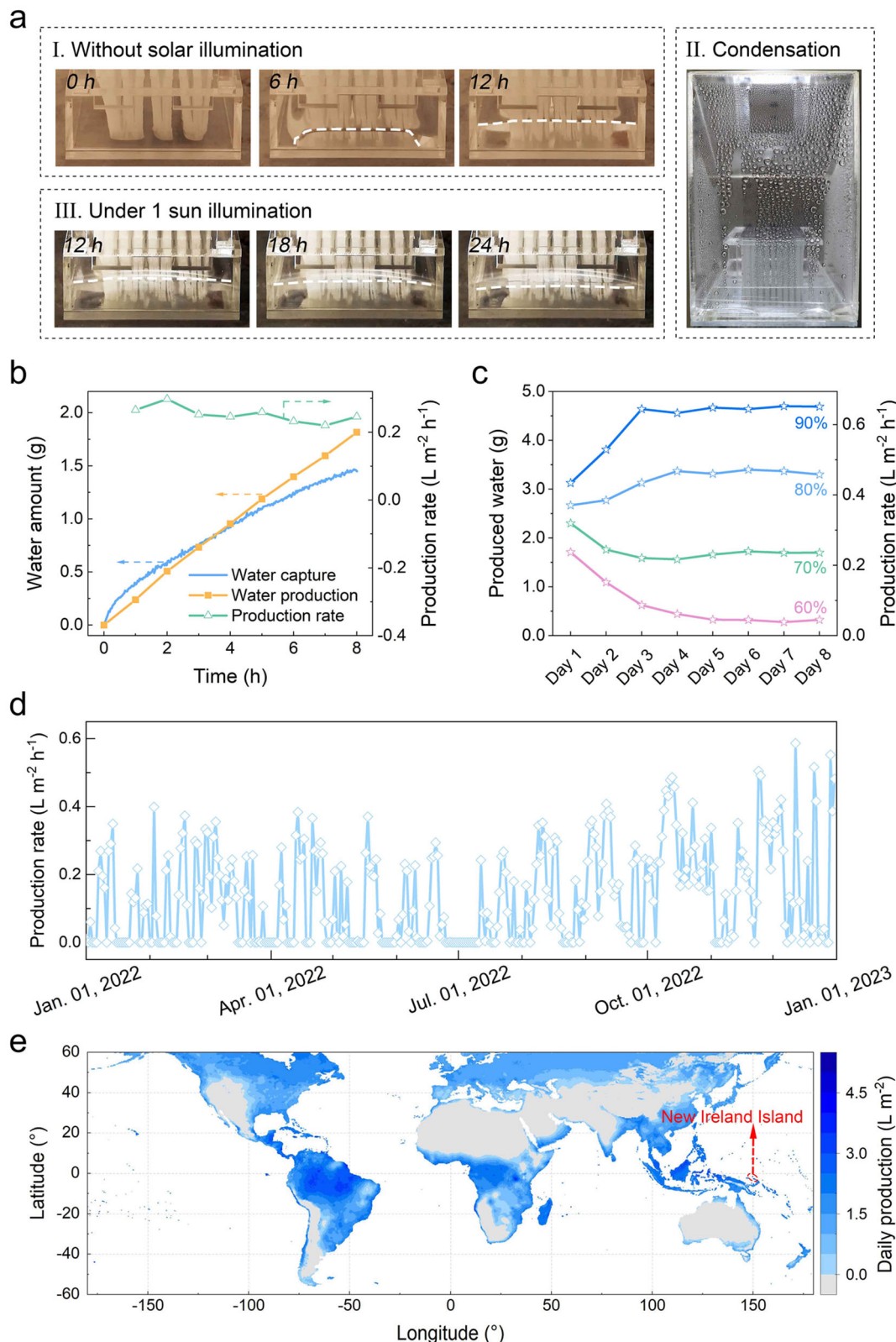

**Fig. 3 | Performance evaluation under controlled environments. a** Snapshots from a video recording the prototype operation process at 90% RH and 25 °C. The prototype was subjected to 12 h of darkness (I), followed by 12 h of 1-sun illumination (III). The white dashed lines are used to guide the viewer's eye to identify the liquid levels in the container since water is transparent and difficult to see. In panel (II), a photo shows water condensation on the chamber's surface. **b** Time-dependent variations in the amount of water captured and produced by the prototype as well as the corresponding calculated water production rates. **c** Water production performance of the prototype under varying RH conditions in maintenance-free mode. **d** Water production variation of the system operated in maintenance-free mode in Jeddah, Saudi Arabia, throughout 2022. **e** A map visualizing the potential of the proposed SAWE system for water production in different regions, calculated based on the global distribution of yearly average solar irradiation, RH, and the water production capability of the prototype. The highlighted area represents New Ireland Island, where the water production can reach up to 4.6 L m⁻² per day.

compensate for the reduced water content in the vapor generation zone, correspondingly decreasing the water level within the container (Fig. 3a-III). Intriguingly, the decrease in the water level was not significant, as the prototype continued to capture water during the water production process. The concurrent processes of water capture and production were both regulated by the RH. Specifically, under 1-sun illumination for 8 h, the prototype produced ~1.8 ml of water while capturing ~1.5 g of atmospheric water from the environment at 65% RH. This corresponds to a water production rate of around 0.22 L m$^{-2}$ h$^{-1}$ (Fig. 3b).

Then, the prototype's performance was comprehensively evaluated under varying RH conditions in an 8-day consecutive water production test (Fig. 3c). Each day of the evaluation comprised 16 h of darkness, followed by 8 h of exposure to 1-sun illumination. The water production rate initially decreased under 60% and 70% RH conditions, yet increased during the initial period under 80% and 90% RH conditions. This variation underscores the system's adaptability to environmental conditions. Throughout this operation, the system achieved a state of mass transport equilibrium and establishing a salt concentration gradient within the MTBs structure that gradually decreased from top to bottom (Supplementary Notes 2 and Supplementary Figs. 21–23). This concentration gradient facilitated automatic sorbent backflow via diffusion and convection, enabling the prototype system to function autonomously without requiring maintenance or adjustments to switch between water capture and water production modes (Supplementary Fig. 24). As the RH increased from 60% to 90%, the stabilized water production rate under 1-sun illumination increased from ~0.04 to ~0.65 kg m$^{-2}$ h$^{-1}$, corresponding to a remarkable increase in solar-water collection efficiency from 3.5% to 44.3% (Supplementary Notes 3 and Supplementary Figs. 25 and 26). These results underscore the capability of the proposed system to extract fresh water from relatively humid air in a maintenance-free mode. Compared with previously reported SAWE devices that often require manual operations or rely on electrical power (see the summarized performance in Supplementary Table 1), the proposed system offers significant advantages in terms of operational simplicity and practicality. To demonstrate the superior performance of our system over a previously reported milestone[26], which was characterized by simultaneous water capture and generation using a high-performance but expensive hygroscopic sorbent ([EMIM][Ac]), we replicated a system according to the documented procedure (see technical details in Supplementary Notes 7). Using the same affordable sorbent (i.e., LiCl), our system exhibited a 9.6-fold increase in water production performance. This significant improvement was achieved through a reduction in system footprint, efficient mass transport, and optimized heat management. This comparison clearly underscores the advancements our system offers in terms of efficiency and cost-effectiveness. Notably, compared with the passive radiative cooling-induced atmospheric water extraction technology[29,30], which ensures uninterrupted water production, the water production rate of this system exceeded by more than 11 times under the same conditions (Supplementary Table 1). Additionally, the system was also assessed in manual mode, during which the chamber was manually opened during the atmospheric water capture process to enhance mass transport (Supplementary Fig. 27). While periodic operation is required in this mode, the system can generate water even at a low RH of 40%, with its water production rate ranging from 0.15 L m$^{-2}$ h$^{-1}$ at 40% RH to 0.68 L m$^{-2}$ h$^{-1}$ at 90% RH (refer to Supplementary Fig. 28). The system outperforms in this mode compared with the maintenance-free mode, excelling in both the applicable RH range and the water production capability at specific RH levels.

By establishing a correlation between the water production rate and RH conditions (Supplementary Fig. 29), the system's year-round water production capacity in Jeddah, Saudi Arabia, was analyzed

(a developed yet water-stressed city with a population of over 4.8 million) based on the daily average RH variations in 2022 (Supplementary Fig. 30). In the maintenance-free operation mode, the system was feasible for more than half of the year and the highest water production rate reached ~0.56 L m$^{-2}$ h$^{-1}$ (Fig. 3d). In manual mode (Supplementary Fig. 31), the system remained functional almost year-round, experiencing only 12 days of unfavorable RH conditions for water production. This underscored the significant potential of the proposed system for SAWE in water-stressed areas.

To further evaluate the global applicability of the proposed system, we created a map illustrating the estimated water production efficiency across the world (Fig. 3e) using global statistics for yearly average RH and solar irradiation (Supplementary Fig. 32)[31,32]. The map reveals that the maintenance-free operation mode of the system is highly suitable for equatorial regions with abundant solar irradiation and high humidity, aligning with the reported global potential for SAWE[1]. Notably, in New Ireland Island, Papua New Guinea, characterized by high RH values and ample solar energy, water production can reach up to 4.6 L m$^{-2}$ per day. In manual mode, the system finds applicability in most regions across the world (Supplementary Fig. 33).

## Field tests

To validate the real-world application and utility of the proposed SAWE system, a larger prototype (evaporation area: 13.5 cm × 24 cm) (Supplementary Fig. 34) was fabricated for outdoor experiment. The experiment was conducted on a rooftop at KAUST (see the experimental setup in Fig. 4a), which commenced on July 27, 2022, at 19:00. After 24 h of operation, the weight of the water produced was measured at 19:00 the following day (see Supplementary Fig. 35 for real-time weather conditions). As shown in Supplementary Movie 2, when the system was exposed to sunlight during the day, ~95 ml of water was generated. Of this, ~85 ml flowed into the graduated cylinder and the remaining 10 ml was retained in the chamber (Fig. 4b). Based on the projected area of the prototype, the water production was calculated to be ~2.9 L m$^{-2}$ per day.

The water production capability of the system was further evaluated over 35 days across two seasons in Thuwal, Saudi Arabia: 10 days in summer and 25 days in fall. With intense sunlight and high temperatures during summer, the daily water production ranged between 65 and 96 ml (equivalent to 2.0–3.0 L m$^{-2}$ per day) (Fig. 4c). The specific daily water production was influenced by both the received solar energy and RH conditions (see real-time weather variations in Supplementary Fig. 35). In the fall, solar intensity and temperature both decreased; however, the prototype remained functional, with daily water production varying between 35 and 90 ml (equivalent to 1.1–2.8 L m$^{-2}$ per day) (Fig. 4c). More intriguingly, due to its completely passive working principle, this system operated spontaneously and continuously to produce water without requiring additional maintenance. Furthermore, the water production can be increased in practical applications by connecting prototypes to meet larger water demands. As shown in Supplementary Fig. 36, with six prototypes connected in series with the total footprint of 23 cm × 93 cm, the scaled-up system produced ~480 ml of freshwater in one day. This value is approximately six times the productivity of each unit.

Notably, as the SAWE system produced water by extracting it from the air and remained exposed to the environment during its entire operation. The collected water may be susceptible to contamination from airborne pollutants. This issue has been observed in other atmospheric water extraction technologies such as fog collection and dew condensation[33,34]. Therefore, the quality of water produced by the system was assessed by determining the concentrations of ions and microbial cells in it. As shown in Fig. 4d, all ionic indicators are well below the guideline values of WHO (indicated by red dashed lines)[35]. For microbiology (Supplementary Table 2), all values of HPC 36oC,

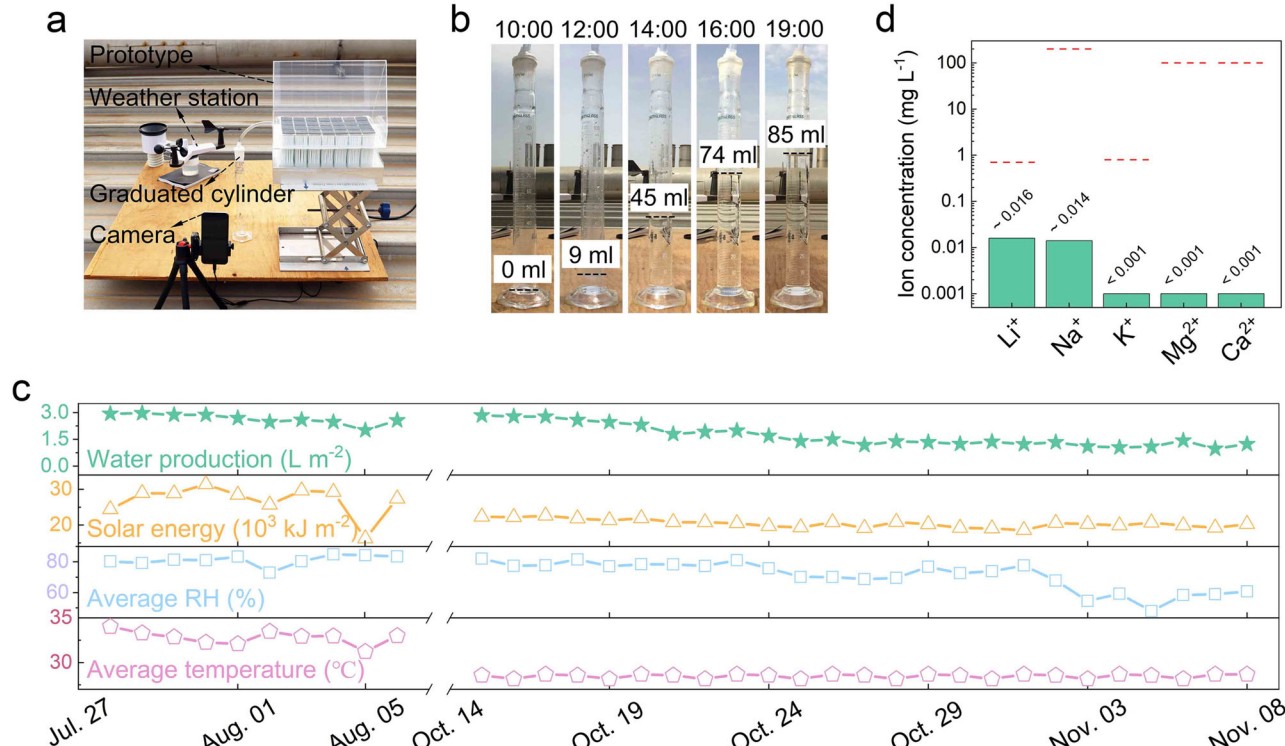

**Fig. 4 | Outdoor performance evaluation. a** Experimental setup, including a weather station for monitoring environmental conditions, a camera for process recording, and a graduated cylinder for condensation collection. **b** Snapshots from a video recording of the practical water production process on July 28, 2022, with the recording starting at 10:00 and ending at 19:00. During this period, approximately 85 ml of water was collected in the graduated cylinder. **c** Water production performance of the prototype and the corresponding weather conditions during a 35-day field test, including a 10-day period from July 27, 2022 to August 6, 2022 and a 25-day period from October 14, 2022 to November 8, 2022. **d** Cation concentration in the collected water, with red dotted lines indicating the drinking water guideline for each indicator.

total coliforms, and *Escherichia coli* were below the detection limits. The measure for active biomass was below the detection limit of the method ($\leq 0.01$ pg ATP mL$^{-1}$), indicating very low or no bacterial presence. This indicates that the water quality met the requirements for drinking and irrigation use.

The SAWE system, which requires no bulk water source and minimal installation and maintenance, enables autonomous, point-of-use irrigation. This represents a revolutionary solution for irrigation in water-stressed regions that lack liquid water resources, as shown in Fig. 5a. To validate this working mode, an automatic drip irrigation system was developed (Fig. 5b). The water from the SAWE system was collected and stored for subsequent drip irrigation at night to avoid excessive evaporation loss during the day. Chinese cabbage was chosen as a representative for growing plants because it is a commercial vegetable sensitive to soil water content. Seeds were double-sowed in a custom-built acrylic tray with 10 individual compartments filled with standard potting soil. Approximately 9 ml of water collected from the SAWE system was delivered to each compartment in the tray at 20:00 each day, totaling 90 ml for daily irrigation application to the growing system.

To compare the use of atmospheric water to conventional water supplies for irrigation applications, an identical growing system supplied with the same volume (~9 ml) of KAUST tap water was built. A third growing system was also tested in which no irrigation water was provided. The volumetric water content (VWC) and electrical conductivity (EC) of the soil were measured daily at 19:00. The soil receiving atmospheric water or tap water irrigation demonstrated comparable VWC and EC of 12% and 145 μS m$^{-1}$, respectively (Fig. 5c) and the plants showed consistent growth (Fig. 5d and Supplementary Figs. 37 and 38, see the height and leaf size variation

recording in Supplementary Fig. 39 and real-time weather condition recording in Supplementary Fig. 40). The system without irrigation showed no signs of plant growth (Supplementary Fig. 41). After 20 days of irrigation, all plants were harvested and their fresh and dry biomass were measured. As shown in Fig. 5e, the biomass of plants irrigated with tap water was similar to those irrigated with collected atmospheric water (see raw data in Supplementary Table 3). To validate the reliability of the reported findings, a repetition experiment involving plant irrigation was conducted on the rooftop in KAUST. The consistent result confirmed the feasibility of utilizing atmospheric water for irrigation (Supplementary Figs. 42 and 43).

An alternative irrigation operation mode was also explored, where a prototype was installed beside each plant pot and the water generated by the prototype flowed directly to the plant's roots (Supplementary Fig. 44). This irrigation mode offered a scalable automatic SAWE irrigation application. The irrigable area that the system could sustain depended on environmental conditions and the specific water requirements of the target plants. To reveal the potential of the developed system for irrigation, two different types of plant growth systems were selected: a cropping system with Chinese cabbage and a desert tree plantation with *Vachellia tortilis* (*Acacia tortilis*). The sustainable irrigation capacity of the system was evaluated based on their daily water requirements (Supplementary Notes 4 and Supplementary Figs. 45 and 46). Specifically, at 70% RH, a 1-m$^2$ system could sustain the growth of up to ~0.8 and ~1.0 m$^2$ of Chinese cabbage in maintenance-free and manual modes, respectively. For desert plantation, under the same conditions, a 1-m$^2$ system could sustain the growth of more than three and four *Vachellia tortilis* in maintenance-free and manual modes, respectively. These

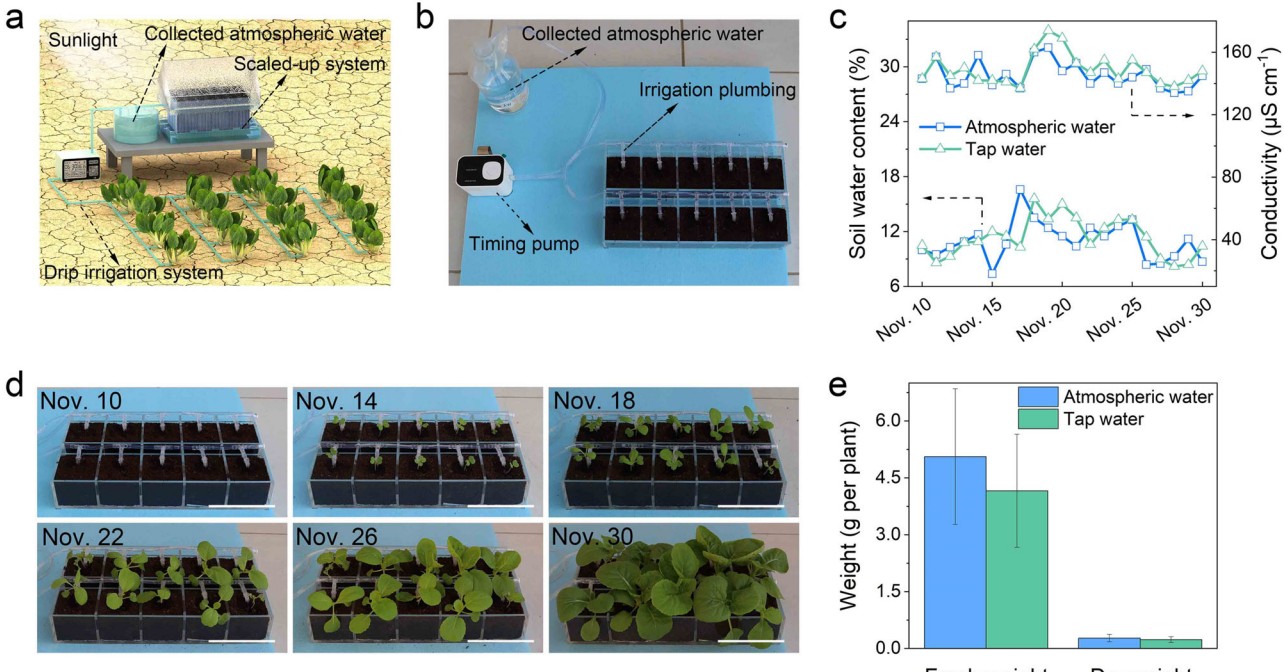

**Fig. 5 | Irrigation using water extracted from the air. a** A conceptual illustration of off-grid irrigation by extracting water from the air. **b** Experiment setup for drip irrigation using the collected atmospheric water that was supplied to the plants by a timing pump via irrigation plumbing. **c** Recording of soil water content and conductivity after irrigation with collected atmospheric water and tap water, from Nov. 10, 2022 to Nov. 30, 2022. **d** Growth progression of plants irrigated with atmospheric water (the scale bar is 10 cm, see Supplementary Fig. 37 for the growth progression of plants irrigated with tap water). **e** Fresh and dry weights of harvested plants, demonstrating comparable growth between plants irrigated with atmospheric water and those irrigated with tap water. Standard deviation presents the weight difference between 10 plants.

results confirm the potential of the proposed SAWE technology in point-of-use irrigation applications for reducing the reliance on ground water resources through harvesting water from the air, particularly in water-scarce regions.

We presented a rationally designed SAWE system for high-performance freshwater production and off-grid irrigation. Unlike conventional atmospheric water–harvesting systems, the proposed prototype eliminates the need for complex system designs and laborious operations. The key to implementing this design lies in leveraging solar-driven transpiration and the resulting concentration gradient to facilitate spontaneous equilibrium between water and sorbent within the MTBs structure. By optimizing the MTBs structure, the prototype produced freshwater at high rates under varying RH conditions. In particular, the system achieved a stabilized water production rate of 0.65 L m$^{-2}$ h$^{-1}$ under 90% RH and 1-sun illumination in the maintenance-free operation mode. In manual mode, the system functioned in environments with RH levels as low as 40%, making it suitable for use in most climates. The practical performance of the system was validated across two seasons in Thuwal, Saudi Arabia. Results indicated its potential for irrigation applications using the collected atmospheric water to grow Chinese cabbage. The off-grid and low-maintenance extraction of atmospheric water that can be supplied directly to plants can revolutionize irrigation in remote, water-scarce regions.

## Methods
### Prototype fabrication
The solar absorber was fabricated by loading partially oxidized CNTs onto the GFM with a controlled loading percentage ~10 wt.%. The MTBs structure was then created by assembling the GFM (~0.45 mm thick with ~60% porosity) into the designed PMMA frame. Specifically, GFM was firstly cut into strips with a width of 3 cm, and the PMMA frame was fabricated via laser cutting and reassembling of the units. Then, these strips were assembled into the designed PMMA frame to realize the MTBs structure fabrication. The final prototype was constructed by combing the MTBs structure, the solar absorber, the condenser chamber and the container together and infiltrating the LiCl solution into the MTBs structure. For other details, please refer to the Supplementary Methods.

### Structure characterization
The microstructure was characterized by SEM (Teneo VS, FEI). The light absorption spectrum of the solar absorber was measured by UV/Vis/NIR spectrometer (Lambda 950, PerkinElmer). The water capture and release isotherms of saturated LiCl solution were evaluated by a dynamic vapor sorption analyzer (IGAsorp, Hiden Isochema). The temperature distribution along the MTBs structure was recorded by IR camera (H16, Hikvision) and the multichannel thermometer (JK808, JinKo) equipped with ultrathin thermocouples.

### Performance evaluation
The performances of prototypes were evaluated in a research chamber (AR66L, Percival) where the temperature and RH can be controlled accurately. The outdoor experiment was conducted on the rooftop in KAUST. The environmental conditions, including RH, temperature, and solar intensity were recorded by a weather station (HP2550, Misol). The off-grid irrigation experiment was performed on the balcony in KAUST. A custom-built acrylic tray with ten individual compartments was used for plant growth. Chinese cabbage seed (Quality Cabbage, Longda Seed) was selected for demonstration and standard potting soil (Basissubstrat 2, Stender) was used for plant growth. For details, please refer to the Supplementary Methods.

**Reporting summary**

Further information on research design is available in the Nature Portfolio Reporting Summary linked to this article.

## Data availability

The authors declare that all data supporting this work are contained in graphics displayed in the main text or in the Supplementary Information. Data used to generate these figures are available from the authors upon request. Source data are provided with this paper.

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

## Acknowledgements

This work is supported by KAUST baseline (BAS/1/1415-01-01 and BAS/1/1372-01-01), KAUST Research Translation Grant (REI/1/5412-01-01), KAUST Competitive Research Grant (URF/1/5019-01-01) and FutureWei's Gift Fund to KAUST (GIF/5/5705-01).

## Author contributions

K.Y. and T.P. conceived the concept, executed the project and authored the manuscript. N.F., A.I.F., P.H., and J.S.V. performed the water quality measurement. R.E.W. provided suggestions for irrigation. Q.G. and Y.H. conceived the concept, supervised this project, and revised the manuscript.

## Competing interests

Patent disclosure based on the result of this work has been submitted through KAUST (Inventor: Yu Han; Kaijie Yang; Tingting Pan; Qiaoqiang Gan. U.S. Provisional Application No. 63/342,781). The remaining authors declare no competing interests.
