## [Peer Review File · Nature Communications]

A solar-driven atmospheric water extractor for off-grid freshwater generation and irrigationReviewers' comments:

Reviewer #1 (Remarks to the Author):

The authors present a comprehensive and original work on a system enabling continuous, autonomous, energy-passive water harvesting, which is evaluated in detail and successfully applied to a relevant use for irrigation of a plant.

A key element of the system is a solar-driven transpiration mechanism, which allows continuous mass transport of water from cold region (where it is collected from the surrounding air by condensation/absorption) and the hot region (heated by exposure to solar light leading to enhanced evaporation). The collection and water transport mechanism involve subtle functions of a water sorbent, which plays a crucial role in water transport (taking advantage of concentration gradient and temperature gradient) in addition to its primary role of enabling efficient water collection.

For all those reasons I recommend considering this paper for possible publication in Nature Communications after revision taking into account the following recommendations and corrections:

(1) page 5, line 135, "the water level in the container gradually rose", when looking to the video, it seems that the water capture is not continuous but reveals sudden increase steps. are these steps related to important variations of the ambient humidity (RH) Can you comment on that?

(2) page 5, line 148, correct this text portion "achieving a state of a state of mass transport equilibrium" by "achieving a state of mass transport equilibrium"

(3) In the Material and Methods section, some important details are missing so as to enable reproducing the results. In particular, the glass fiber membrane (GFM), supplier and references/specifications is required. Same for other chemicals (CNTs, LiCl).

Also please clarify how the GMF sheets are processed to form the GMF arrayed structure described in Figure 1a.

(4) In Supplementary Fig.2, The fact that MB appears in the photo suggests that the GFM are (partially) translucent to visible light. Can you confirm this fact ? Is there any possible residual impact of this partial translucency to induce parasitic heating through lateral solar irradiation of the system?

(5) In Supplementary Fig.19, for better readability, please add the texts "30% RH" ; "60% RH." , "90% RH.", in Figures a, b and c, respectively

(6) In Supplementary Fig.26, how this efficiency is defined ? Is it the ratio of total latent heat of evaporation over incident solar energy?

(7) Reference is needed for the source of the map shown in Supplementary Fig.32

(8) Supplementary Table 1. I recommend complementing this table (and your reference lists) based on additional benchmarking information that you can find in these two recent review papers on energy-passive atmospheric water harvesting:

X. Liu et al. ACS Materials Letters 4 (5), 1003-1024, 2022

A. Entezari et al. Advanced Materials 35, 2210957, 2023

Reviewer #2 (Remarks to the Author):

The work, “A solar-driven atmospheric water extractor for off-grid freshwater generation and irrigation” introduces a solar-driven atmospheric water extraction (SAWE) system and its potential application in harvesting water from the air for agricultural irrigation. This SAWE system is made from economic glass fiber membranes (GFMs), CNTs (as the solar light absorbers), and hygroscopic LiCl salts. By simply confining CNTs on one end of the membrane, a temperature gradient, a salt concentration gradient, and a division of two functional regions (including the high T vapor generation zone and the low T water capture zone) are created along the membranes. This asymmetric structure helps facilitate a continuous and constantly-looped water absorption-transportation-evaporation-condensation process to autonomously generate freshwater from the air.

This work attempts to show that the SAWE system developed is a simple and effective approach to generating freshwater for irrigation. However, the manuscript may not be considered for publication as there are many concerns as follows:

1. What is the difference between the SAWE system developed in this work and other SAWE systems? The novelty is not clear. The SAWH system developed here seems to work in a night-absorption-and-daytime-desorption mode, similar to other SAWE systems, for example, Qi H, Wei T, Zhao W, et al. An interfacial solar-driven atmospheric water generator based on a liquid sorbent with simultaneous adsorption–desorption[J]. Advanced Materials, 2019, 31(43): 1903378. Wang X, Li X, Liu G, et al. An interfacial solar heating assisted liquid sorbent atmospheric water generator[J]. Angewandte Chemie, 2019, 131(35): 12182-12186.

Moreover, the water collection rate are lower than the reported work.

2. Have the dependence of water production rate on the RH and solar irradiation been quantified? How are the estimations of the daily water production on a global scale done (Figure 3d, 3e, Figure S33)? To explain clearly.

3. The data presentation should be improved. For example, using snapshots to show the change in water amount (Figure 3a, 4b) is not an accurate and clear method.

4. Why does the water production under 60%RH, 70%RH, and 80%RH reduce for the first three days?

5. What are the criteria for optimizing the height of the vapor generation zone and the water capture zone, especially the water capture zone? “The water capture rate is proportional to the H_a , which significantly increased from 0.6 kg/m²/h at the height of 1 cm to ~6.2 kg/m²/h at the height of 9 cm” (Line 94-95, SI), why the optimized H_a is 5 cm?

6. Does the capillary force affect the backflow of the highly concentrated salt solution? If yes, the equation 1 (SI) may be incorrect.
7. Why does the evaporation rate of the prototype with 2 GFM decrease during the operation? To explain the effect of the number of GFMs on the evaporation rate?
8. The labels in Figure S19b are wrong.
9. What does the yellow curve in Fig S20 mean?
10. The organization and writing of this manuscript can be much improved.

Point-by-Point Responses to the Reviewers' Comments

Reviewer: 1

General Comments: The authors present a comprehensive and original work on a system enabling continuous, autonomous, energy-passive water harvesting, which is evaluated in detail and successfully applied to a relevant use for irrigation of a plant.

A key element of the system is a solar-driven transpiration mechanism, which allows continuous mass transport of water from cold region (where it is collected from the surrounding air by condensation/absorption) and the hot region (heated by exposure to solar light leading to enhanced evaporation). The collection and water transport mechanism involve subtle functions of a water sorbent, which plays a crucial role in water transport (taking advantage of concentration gradient and temperature gradient) in addition to its primary role of enabling efficient water collection.

For all those reasons I recommend considering this paper for possible publication in Nature Communications after revision taking into account the following recommendations and corrections:

General responses: We are truly grateful for this reviewer's insightful review and acknowledgment of our work. We have carefully considered all valuable suggestions and made revisions for improvement. Below, you will find our detailed point-by-point responses.

Comment 1: page 5, line 135, "the water level in the container gradually rose", when looking to the video, it seems that the water capture is not continuous but reveals sudden increase steps. are these steps related to important variations of the ambient humidity (RH) Can you comment on that?

Responses: Thank you very much for this insightful comment. Throughout the evaluation process, the relative humidity (RH) was consistently maintained at 60%. The variation in water capture rate is attributed to change in LiCl concentration during operation. As water molecules are captured, the concentration of LiCl decreases, consequently the water capture rate will slow down. We investigated the dependence of water capture rate on the RH and LiCl concentration, and presented the results in Supplementary Fig. 19 and 20. Please refer to this part for details. To improve the accuracy, in this revision, we have revised the sentence to 'the water level in the container increased' (Page 5, line 136).

Comment 2: page 5, line 148, correct this text portion "achieving a state of a state of mass transport equilibrium" by "achieving a state of mass transport equilibrium"

Responses: Thank you very much for your meticulous review. We have made revision to correct the typo (Page 5, line 151).

Comment 3: In the Material and Methods section, some important details are missing so as to enable reproducing the results. In particular, the glass fiber membrane (GFM), supplier and references/specifications is required. Same for other chemicals (CNTs, LiCl).

Also please clarify how the GFM sheets are processed to form the GFM arrayed structure described in Figure 1a.

Responses: Thanks for this suggestion. The GFM used in our system was purchased from a commercial supplier (i.e., Haining Taoyuan Chemical Instrument Factory, with a product model of 49# glass fiber membrane). The CNTs were acquired from Sigma-Aldrich, with specifications indicating a diameter of 110-170 nm and a length of 5-9 μm . LiCl was obtained from VWR Chemicals. To obtain the prototype as described in Fig. 1a, GFM was firstly cut into strips with a width of 3 cm, and the PMMA frame was fabricated via laser cutting and reassembling of the units. Then, these strips were assembled into the designed PMMA frame to realize the MTBs structure fabrication. The final prototype was constructed by combing the MTBs structure, the solar absorber, the condenser chamber and the container together and infiltrating the LiCl solution into the MTBs structure. The detailed procedure was depicted in the Supplementary Fig. 3. In this revision, we have supplemented all details in Supplementary Note 7 ('Material and methods – Prototype fabrication').

Comment 4: In Supplementary Fig.2, The fact that MB appears in the photo suggests that the GFM are (partially) translucent to visible light. Can you confirm this fact? Is there any possible residual impact of this partial translucency to induce parasitic heating through lateral solar irradiation of the system?

Responses: We appreciate your thorough review. The transmittance of GFM is near to 0%, rendering it non-transparent to visible light (see Fig. R1). Additionally, since GFM also exhibits low absorption of sunlight ($\sim 6.7\%$, Fig. R2), the impact of lateral solar irradiation is negligible.

Figure R1. The UV-Vis-NIR spectra of GFM. a, the transmittance spectrum of GFM, its transmittance towards sunlight is near 0, indicating its opacity to solar radiation. **b,** the absorption spectrum of GFM, the calculated absorption to sunlight is $\sim 6.7\%$.

Comment 5: In Supplementary Fig.19, for better readability, please add the texts "30% RH" ; "60% RH." , "90% RH." , in Figures a, b and c, respectively

Responses: We appreciate this constructive suggestion. In this revision, we have added the RH condition to increase the readability. Please see the revised Supplementary Fig. 19.

Comment 6: In Supplementary Fig.26, how this efficiency is defined ? Is it the ratio of total latent heat of evaporation over incident solar energy?

Responses: The solar-water collection efficiency was calculated using the following equation: $\eta_s = (h_{lv} + h_d)m_{cw}/A_{sa}q_{sol}t$, where η_s presents the solar-water collection efficiency, h_{lv} is the evaporation enthalpy of pure water, h_d is the differential enthalpy of dilution, m_{cw} is the collected water amount, A_{sa} is the size of the solar absorber, q_{sol} is the solar flux and t is the operation time. As the reviewer can see, it is the ratio of the required heat energy for evaporation to the incoming solar energy. This method was also used in *Energy Environ. Sci.*, **11**, 1510-1519, (2018) and *Angew. Chem. Int. Ed.* **58**, 12054-12058, (2019) (cited as ref. 20). The detailed calculation procedure of the solar-water collection efficiency was described in Supplementary Note 3 (i.e., ‘Solar-water collection efficiency calculation’).

Comment 7: Reference is needed for the source of the map shown in Supplementary Fig.32

Responses: The yearly average RH and solar irradiation were cited from data source of “*Global Solar Atlas*” (ref. 31) and “*Center for Sustainability and the Global Environment*” (ref. 32), respectively. These two data sources were cited as ref. 31 and 32 in the manuscript.

Comment 8: Supplementary Table 1. I recommend complementing this table (and your reference lists) based on additional benchmarking information that you can find in these two recent review papers on energy-passive atmospheric water harvesting:

X. Liu et al. *ACS Materials Letters* 4 (5), 1003-1024, 2022

A. Entezari et al. *Advanced Materials* 35, 2210957, 2023

Responses: We appreciate this comment for comprehensiveness improvement. These two studies have been carefully reviewed and two pioneering systems have been incorporated in the Supplementary Table 1 for quality improvement.

Reviewer: 2

General Comments: The work, “A solar-driven atmospheric water extractor for off-grid freshwater generation and irrigation” introduces a solar-driven atmospheric water extraction (SAWE) system and its potential application in harvesting water from the air for agricultural irrigation. This SAWE system is made from economic glass fiber membranes (GFMs), CNTs (as the solar light absorbers), and hygroscopic LiCl salts. By simply confining CNTs on one end of the membrane, a temperature gradient, a salt concentration gradient, and a division of two functional regions (including the high T vapor generation zone and the low T water capture zone) are created along the membranes. This asymmetric structure helps facilitate a continuous and constantly-looped water absorption-transportation-evaporation-condensation process to autonomously generate freshwater from the air.

This work attempts to show that the SAWE system developed is a simple and effective approach to generating freshwater for irrigation. However, the manuscript may not be considered for publication as there are many concerns as follows:

General Responses: Thank you for reviewing our manuscript and for your valuable insights aimed at enhancing its quality and elucidating its innovative aspects. We deeply appreciate your recognition of the simplicity and effectiveness of our system for continuous and automatic atmospheric water harvesting.

Given the multidisciplinary nature of our research, we have endeavored to cover a broad range of topics, including system design and optimization for maintenance-free operation, as well as its unique application in direct point-of-use irrigation. In this revision, we wish to respectfully highlight a key aspect of our work that may have been overlooked.

Firstly, our system is engineered to directly supply atmospheric water to support plant growth in water-stressed regions such as Jeddah, Saudi Arabia. Given the critical global challenge of freshwater scarcity, our off-grid, low-maintenance solution for extracting atmospheric water and delivering it directly to plants represents a transformative innovation for irrigation in remote, arid areas.

Secondly, atmospheric water harvesting provides a sustainable approach for freshwater production. For example, the use of Metal-Organic Framework (MOF) materials for atmospheric water extraction has been heralded as a significant advancement for off-grid water production, as reported in *Science* **356**, 430–434, (2017). However, most existing implementations still necessitate extensive maintenance to collect fresh water (e.g., *Nat. Commun.* **9**, 1191, (2018); *Angew. Chem. Int. Ed.* **58**, 12054-12058, (2019); *Nat. Nanotech.* **17**, 857-863, (2022)). To overcome this practical limitation and explore a more promising application in humid environments, where more water vapor is present in the air, we have developed the current portable and practical system that autonomously produces water from the air. We will provide further details in our point-by-point responses to your specific questions below.

Additionally, in response to your constructive feedback, we have meticulously enhanced the quality of our manuscript and further underscored its innovative contributions. We have also engaged a professional language editing service to ensure thorough refinement of the text. We trust these revisions comprehensively address your concerns and invite you to reconsider our work for publication.

Comment 1: What is the difference between the SAWE system developed in this work and other SAWE systems? The novelty is not clear. The SAWH system developed here seems to work in a night-absorption-and-daytime-desorption mode, similar to other SAWE systems, for example,

Qi H, Wei T, Zhao W, et al. An interfacial solar - driven atmospheric water generator based on a liquid sorbent with simultaneous adsorption - desorption[J]. *Advanced Materials*, 2019, 31(43): 1903378.

Wang X, Li X, Liu G, et al. An interfacial solar heating assisted liquid sorbent atmospheric water generator[J]. *Angewandte Chemie*, 2019, 131(35): 12182-12186.

Moreover, the water collection rate are lower than the reported work.

Responses: We sincerely appreciate your comment asking us to clarify the novelty of our current work compared with existing literature. The two pioneering studies mentioned are indeed acknowledged in our original manuscript, cited as ref. 20 and ref. 26, respectively. We have fully recognized these prior works.

Our system is distinct from the earlier systems in several critical aspects including design, working principles, and applications, which will be detailed subsequently. Importantly, thanks to the innovative design of the MTBs structure in our system, it not only enables simultaneous water capture and generation under sunlight but also ensures efficient heat management and effective mass transport. This rationally designed system significantly minimized the footprint of the system and increased the water production performance up to 9.6 times higher than the pioneering system reported in *Adv. Mater.* **31**, e1903378, (2019) when using same liquid sorbent (i.e. LiCl). This underscores the clear advantage of our system design.

To clarify further, here are the specific differences from the two pioneering works:

- A. In the study published in *Angew. Chem. Int. Ed.* **58**, 12054-12058, (2019), the authors pioneered the use of interfacial solar heating to aid atmospheric water harvesting by integrating a graphene oxide (GO)-based solar absorber on the surface of a bulk CaCl_2 solution. However, this system requires manual intervention to open and close the chamber for operational switching, making it labor-intensive. Additionally, due to the slow water capture rate, it is constrained to a daily cycle, with water capture occurring at night and water generation during the daytime. In contrast, the proposed system in this study well addressed these challenges. It not only realized simultaneous atmospheric water capture and generation under sunlight, liberating itself from daily cycle, but also can operate automatically without the need for additional labor input.
- B. *Adv. Mater.* **31**, e1903378, (2019) introduced a novel system for atmospheric water harvesting that features simultaneous adsorption-desorption capabilities, marking a significant advancement in the field. However, this system predominantly focused on interfacial evaporation using a notably expensive hygroscopic sorbent, [EMIM][Ac], priced at approximately \$35 per gram. In light of the practical cost considerations, we opted for a more economically viable sorbent, LiCl, which costs only \$0.7 per gram—just 2% of the cost of [EMIM][Ac].

Moreover, the system described in this pioneering *Adv. Mater.* paper encountered several challenges: (1) high heat dissipation—placing the solar absorber directly on the liquid sorbent surface led to

considerable heat loss into the bulk solution, compromising efficiency; (2) extensive footprint requirement—the design required a large water capture area around the evaporation region, significantly increasing the system's footprint and reducing water production per unit area. These issues substantially limited its practical performance in atmospheric water extraction (AWE). To address these drawbacks, our work introduces an optimized system featuring efficient mass transport and effective heat management through a strategically designed MTBs structure. This configuration has demonstrated a dramatic increase in performance, achieving water production rates up to 9.6 times higher than the aforementioned system, using the same LiCl solution (see the new experimental data in Figs. R2 and R3 below). Further, we have explored the practicality of using the harvested water for direct agricultural irrigation, proposing new sustainable approaches within the water-energy-food nexus.

Next, we will present a comprehensive overview of the significant progress achieved beyond the pioneering efforts, detailing how our innovations have enhanced both the efficiency and applicability of atmospheric water harvesting technologies.

1. Revolutionary Off-Grid Irrigation for Sustainable Agriculture: Our study introduces an innovative approach to addressing off-grid irrigation needs through the extraction of atmospheric water. The immense potential of this approach is substantiated by our meticulously engineered system, poised to redefine direct irrigation practices in remote, water-scarce regions—a feat, to our knowledge, never before reported. Our research extends its impact into realms of sustainability beyond those covered by previously reported works. This type of off-grid irrigation is crucial for ambitious sustainability initiatives, such as the Saudi Green Initiative, which aims to plant 10 billion trees in designated areas (<https://www.greeninitiatives.gov.sa/about-sgi/sgi-targets/greening-saudi/>). We hope that the interdisciplinary potential of our innovation will be assessed on a broader scale, illuminating dimensions not previously explored in foundational research.

2. A rationally designed architecture for optimized heat management and high-performance water production: The key feature of the system explored in this study lies in its innovative MTBs, seamlessly integrating two distinct functional zones within the vertical space. This meticulously engineered architectural design serves a dual purpose, facilitating not only the efficient extraction of water but also enabling the effective sorbent transport and heat recuperation (as elucidated in the part of 'MTBs structure optimization' in Supplementary Note 1). This strategic integration ensures exceptional water production capabilities and maintains outstanding performance stability. Consequently, our system demonstrates a substantial improvement in water production performance. In order to provide an impartial performance comparison and highlight the inherent advantages of our architectural design in freshwater production, in this revision, we reproduced the system detailed in *Adv. Mater.* **31**, e1903378, (2019) (Fig. R2a) and conducted a rigorous evaluation under identical conditions, where the system's dimensions were meticulously tailored in accordance with the size specifications provided in the referenced literature (Fig. R2b). Specifically, this evaluation was carried out consistently at 70% RH and 25°C, utilizing a LiCl solution with an initial concentration of 0.24 g/g as the liquid sorbent. The evaluation lasted 8 days, with 8 hours under 1 sun illumination and 16 hours

under dark environment. As illustrated in Fig. R3, normalized by the footprint of the system, the water production of our system is around 9.6 times higher than the pioneering system reported *Adv. Mater.* **31**, e1903378, (2019). This striking difference demonstrates a remarkable improvement in water productivity, attributable to the efficient heat energy utilization ('MTBs structure optimization – Optimization of Vapor Generation Zone' in Supplementary Note 1) and meticulous optimization of architecture that significantly reduce the system's physical footprint. To provide additional clarity, when aiming for a target water productivity of 5 kg/day, our system would necessitate a mere 2.6 m², whereas the control system would require a much more substantial 25 m². This underscores the unmistakable advantage of our innovative design, not only in terms of performance but also in optimizing space efficiency. These attributes are of paramount importance for the practical implementation of such an AWE device, as they maximize its potential impact on sustainability efforts. To clearly illustrate the advantage and the enhancement of the proposed system, we have incorporated this comparison into the revised manuscript. Please kindly refer to Page 6, line 163-171 and Supplementary Note 8.

Fig. R2. The system reported in the previous study of *Adv. Mater.* **31**, e1903378, (2019). **a**, A photo of the fabricated system. **b**, The detailed size information of the fabricated system

Fig. R3. Performance comparison between the reported system and our system at 70% RH.

Comment 2: Have the dependence of water production rate on the RH and solar irradiation been quantified? How are the estimations of the daily water production on a global scale done (Figure 3d, 3e, Figure S33)? To explain clearly.

Responses: In this study, we thoughtfully investigated the water production capability of system under different RH conditions in both maintenance-free mode and manual mode under 1 sun illumination. Please kindly refer to Fig. 3c and Supplementary Fig. 28.

The year-round water production estimation for Jeddah, Saudi Arabia (Fig. 3d) was calculated by considering the solar-water collection efficiency across different RH conditions, alongside the daily average humidity and solar irradiation data for Jeddah in 2022. While, the water production estimation on a global scale (Fig. 3e and Supplementary Fig. 33) was derived by incorporating the solar-water collection efficiency across varying RH conditions, along with the yearly average humidity and solar irradiation data. In detail, the solar-thermal conversion efficiency under various RH was determined by applying the formula of $\eta_s = \frac{(h_{lv} + h_d) \times m_{cw}}{A_{sa} \times q_{sol} \times t}$, considering the measured water production capability under corresponding RH condition. In turn, based on the historical humidity and solar irradiation data of specific regions, along with the efficiency of the system, we can estimate the water production. To enhance the clarity, we have supplemented the detailed calculation method and the environment data source in the Supplementary Note 3 – Water production estimation.

Comment 3: The data presentation should be improved. For example, using snapshots to show the change in water amount (Figure 3a, 4b) is not an accurate and clear method.

Responses: We appreciate this comment. The snapshots and supplementary video provided are designed to give readers visual insights into the operational process of the system (Fig. 3a) and the practical water production (Fig. 4b). They were not intended to quantify the volume of captured or produced water. As outlined in the part of ‘Materials and methods – Performance evaluation’ in Supplementary Note 7, the water capture performance was determined by recording the weight change of the prototype and the water production was measured by weighting the condensed water. The accurately measured data under different conditions are presented in Fig. 3b and 3c, and Supplementary Fig. 20. In this revision, we have further emphasized and clarified this point in the manuscript to prevent any potential misunderstanding (Page 5, line 133-134).

Comment 4: Why does the water production under 60%RH, 70%RH, and 80%RH reduce for the first three days?

Responses: The water production rate initially decreased under 60% and 70% relative humidity (RH) conditions, yet increased during the initial period under 80% and 90% RH conditions. This variation underscores the system's adaptability to environmental conditions. Initially, the MTB structure was saturated with a LiCl solution at a concentration of 0.24 g/g. Under the low RH conditions (60% and 70%),

water molecules release causes the LiCl concentration in the vapor generation zone to increase, resulting in a decreased water production rate. As the concentration stabilizes, the system reaches a mass transport equilibrium and establishes a salt concentration gradient within the MTB structure, which then allows for stable water production. We have revised the description of this dynamic in Page 5, line 148-152 to clarify these processes in the manuscript.

Comment 5: What are the criteria for optimizing the height of the vapor generation zone and the water capture zone, especially the water capture zone? “The water capture rate is proportional to the H_a , which significantly increased from 0.6 kg/m²/h at the height of 1 cm to ~6.2 kg/m²/h at the height of 9 cm” (Line 94-95, SI), why the optimized H_a is 5 cm?

Responses: We appreciate this comment. A detailed analysis of height optimization was provided in the part of ‘MTBs structure optimization – Atmospheric water capture zone’ (Supplementary Note 1), where we have enhanced the explanation. We determined the optimal height (H_a) to be 5 cm, as this height typically ensures that the water capture rate exceeds the water production rate under most conditions. In other words, increasing H_a beyond 5 cm does not improve the water production performance of the system. For these technical details, please refer to Supplementary Note 1.

Comment 6: Does the capillary force affect the backflow of the highly concentrated salt solution? If yes, the equation 1 (SI) may be incorrect.

Responses: We appreciate your insightful inquiry regarding the impact of capillary forces on the backflow of the highly concentrated salt solution. Indeed, capillary forces stemming from GFM’s porous structure are instrumental in elevating the bulk solution. However, once the micro-channels within the GFM are saturated with solution, ions backflow within the bulk solution occurs due to the concentration gradient through diffusion and convection. Thereby, the influence of capillary force on the ions transport within the bulk solution is minimal. This dynamic is well-documented in the literature, including in "Convective Heat and Mass Transfer in Porous Media" (1991, Kluwer Academic Publishers) and "Introduction to Transport Phenomena Modeling" (2017, Springer Nature Switzerland AG). It is established that convection and diffusion serve as the primary mechanisms for backflow during the vapor generation process, as supported by findings in *Energy Environ. Sci.* **11**, 1510 (2018) and *Nat. Comm.* **13**, 849 (2022). Consequently, Equation 1 (SI) in our submission remains valid for analyzing the mechanisms at play in this context.

Comment 7: Why does the evaporation rate of the prototype with 2 GFM decrease during the operation? To explain the effect of the number of GFMs on the evaporation rate?

Responses: We value your thorough scrutiny of the technical specifics. In our experimental setup utilizing two GFMs, a notable decrease in evaporation rate was indeed observed during operation, as detailed in Supplementary Figure 12. Our analysis suggests the following dynamics: As evaporation progresses, water

molecules escape, and the concentration of LiCl solution within the high-temperature zone of the prototype incrementally increases. In setups involving two GFMs, this results in an insufficiency of mass transport channels, which hampers the effective backward transport of accumulated LiCl. Consequently, this heightened salt concentration leads to a diminished evaporation rate.

The number of GFMs within the system is pivotal in managing the sorbent backflow. Our design allows for the concentrated sorbent to be returned through the GFMs via diffusion and convection processes. The mass flow equation, $J = J_{diff} + J_{conv} = nA\varepsilon(k_d\nabla C + k_c\nabla\rho)$, illustrates that the sorbent mass flow is directly proportional to the number of GFM layers (n). Therefore, increasing the GFM number can significantly enhance sorbent backflow by providing more backflow channels, effectively countering the observed reduction in evaporation rate during operation.

To provide a comprehensive understanding of the influence of the number of GFMs on evaporation performance, we conducted systematic investigations as part of this revision. These findings, presented in the 'MTBs Structure Optimization – GFM Sheets Number Optimization' section of Supplementary Note 1, offer crucial insights for optimizing the configuration of GFMs to maximize operational efficiency.

Comment 8: The labels in Figure S19b are wrong.

Responses: Thank you very much for your careful review. We have corrected the typo. Furthermore, we have also added the description of corresponding RH in each figure to provide readers with direct insight into the operation conditions.

Comment 9: What does the yellow curve in Fig S20 mean?

Responses: The yellow curve represents the water production rate of the prototype using LiCl solutions at varying concentrations. We had used a yellow arrow to connect the yellow curve to the vertical line on the right, to illustrate this relationship. To enhance the clarity, in this version we have this detail in the chart's title. Please kindly refer to the Supplementary Fig. 20.

Comment 10: The organization and writing of this manuscript can be much improved.

Responses: We appreciate this helpful comment. After addressing all technical feedback, we have also engaged a professional language editing service to ensure thorough refinement of the manuscript's text.

REVIEWER COMMENTS

Reviewer #1 (Remarks to the Author):

The authors addressed most of my recommendations in a satisfactory manner.

Still it is not understandable that Methylene Blue (MB) can be seen clearly as a blue colour through the GFM in Supplementary Fig. 2, although the authors provided a measured transmittance spectrum which shows that visible light cannot be transmitted by the GFM.

These 2 information seem contradictory and requires some clarification.

Therefore, I recommend a second round of revision to clarify this point, although it is minor, it will avoid any inconsistency in the paper.

Reviewer #2 (Remarks to the Author):

The authors have revised the paper adequately. I have no further input.

Point-by-Point Responses to the Reviewers' Comments

Reviewer: 1

General Comments: The authors addressed most of my recommendations in a satisfactory manner.

Still it is not understandable that Methylene Blue (MB) can be seen clearly as a blue colour through the GFM in Supplementary Fig. 2, although the authors provided a measured transmittance spectrum which shows that visible light cannot be transmitted by the GFM.

These 2 information seem contradictory and requires some clarification.

Therefore, I recommend a second round of revision to clarify this point, although it is minor, it will avoid any inconsistency in the paper.

General responses: We appreciate this reviewer's careful review and meticulous attention to detail. In our experiment, MB solution was used to visually demonstrate the capillary rise process of water in the GFM. The GFM has an intertwined structure (as shown in Fig. 2b in the main text) with a high porosity of ~60%. During the capillary rise process, MB solution fills the porous spaces in the GFM, resulting in a blue appearance. However, this phenomenon is independent of the GFM's opacity. For instance, when a white hydrophilic fabric is immersed in dye solution, it becomes saturated with dye solution and takes on the corresponding color, but it doesn't imply transparency of the fabric. Similarly, the GFM appears blue because the dye fills its internal voids. This phenomenon is not contradictory to the fact that the GFM itself is opaque. To offer readers an explicit understanding, we have added further clarification to underscore this point in Supplementary Fig. 2 (i.e., Fig. R1 below).

Fig. R1 | Capillary rise of water along the GFM. The water was dyed by methylene blue (MB) to enhance the visualization. During the capillary rise process, MB solution fills the porous spaces in the GFM, giving rise to a blue appearance that effectively illustrate the solution transport process.

REVIEWERS' COMMENTS

Reviewer #1 (Remarks to the Author):

I have no more concerns about the manuscript. I recommend accepting this paper as is.